# Potential Use of 3D CORAGRAF-Loaded PDGF-BB in PLGA Microsphere Seeded Mesenchymal Stromal Cells in Enhancing the Repair of Calvaria Critical-Size Bone Defect in Rat Model

**DOI:** 10.3390/md20090561

**Published:** 2022-08-31

**Authors:** Saktiswaren Mohan, Puvanan Karunanithi, Malliga Raman Murali, Khairul Anwar Ayob, Jayaraman Megala, Krishnamurithy Genasan, Tunku Kamarul, Hanumantha Rao Balaji Raghavendran

**Affiliations:** 1National Orthopaedic Centre of Excellence in Research and Learning (NOCERAL), Department of Orthopaedic Surgery, Faculty of Medicine, University of Malaya, Kuala Lumpur 50603, Malaysia; 2Department of Anatomy, Faculty of Medicine, Manipal University College Malaysia, Melaka 75150, Malaysia; 3Department of Genetic Engineering, Faculty of Engineering and Technology, SRM Institute of Science and Technology, SRM Nagar, Kattankulathur, Kanchipuram, Chennai 603203, Tamil Nadu, India; 4Department of Physiology, Faculty of Medicine, University of Malaya, Kuala Lumpur 50603, Malaysia; 5Advanced Medical and Dental Institute (AMDI), University Sains Malaysia, Bertam, Kepala Batas 13200, Malaysia; 6Biomaterials Laboratory, Faculty of Clinical Research, Central Research Facility, Sri Ramachandra Institute of Higher Education and Research, Chennai 600 116, Tamil Nadu, India

**Keywords:** coral, calvaria, platelet-derived growth factor, microsphere, micro-CT, histology

## Abstract

Our previous study evidenced that the 3D CORAGRAF loaded with PLGA microsphere constitutes PDGF-BB can support cell attachment and proliferation and can induce an osteogenic commitment of mesenchymal stromal cells in the in vitro condition. However, how this construct can perform in pathophysiological conditions in terms of repairing critical bone defects is yet to be understood. A study was therefore conducted to investigate the regeneration potential of calvaria critical-size defects using CORAGRAF + PLGA with PDGF-BB + mesenchymal stromal cells (MSCs) in a rat model. A 5 mm critical bone defect was created on calvaria of 40 male Sprague-Dawley rats. CORAGRAF incorporated either with or without PDGF-BB and seeded with rat bone-marrow-derived MSCs was implanted at the defect region. The bone regeneration potential of implanted constructs was assessed using micro-CT imaging and histological staining in weeks 4 and 8. The micro-CT images indicated a significant closure of defects in the cranial bone of the rats treated with 3D CORAGRAF + PLGA with PDGF-BB + MSCs on week 4 and 8 post-implantation. This finding, further supported with the histology outcome where the rat cranial defect treated with CORAGRAF + PLGA with PDGF-BB + MSCs indicated neo-bony ingrowth with organized and mature bone-like morphology as compared with other groups. The previous in vitro results substantiated with our pre-clinical findings demonstrate that the combination of CORAGRAF + PLGA with PDGF-BB + MSCs could be an ideal construct to support bone regeneration in critical bone defects. Hence, this construct can be further investigated for its safety and efficacy in large animal models, or it can be skipped to human trial prior for commercialization.

## 1. Introduction

A successful treatment modality for congenital and craniofacial bone defects remains to be a foremost concern and challenge to surgeons [1]. The demand for enhancing healing in bone defects due to infection, tumour, bone disease, injuries, or congenital malformation increases each year with over millions of cases worldwide [2]. Three main options to repair bones with critical-size defects are: autologous grafts, allografts, and tissue-engineered bone transplantations [3]. Autologous grafts remain as the gold standard and first selection due to their excellent bone regeneration capacity. However, risk of donor site morbidity, infection, and limitations in availability demerit its wide-spread application [4]. The secondary option is allografts, which can provide numerous advantages, such as limitless supply, avoiding donor site morbidity, and decreased surgical time. However, its poor integration time and issues with donor-transmitted diseases to the recipients disadvantage its regular use [5]. Stem cell biology and regenerative medicine, with the support of tissue engineering, provide innovative modalities of treatment for bone augmentation through linking the use of scaffolds, stem cells, and cell signalling molecules [6]. An ideal scaffold should provide a framework for cell adhesion and must possess osteoconductive, osteoinductive, and osteointegration properties [7,8]. 

Synthetic ceramic bone scaffolds, such as hydroxyapatite and beta-tricalcium phosphate (β-TCP), receive significant attention due to their osteoconductive properties [9]. However, the use of these biomaterials is limited due to their inherent characteristics, including rapid resorption rate, brittleness, and difficulty in molding the material [10]. The focus was later channelled to abundantly available marine resources such as coral. It was a remarkable bioresource that has an organically engineered structure which supports human tissue augmentation and regeneration. Many marine products such as seashells, sea urchins, and coral had been reported with unique morphology relevant to bone development, such as interconnected porous and dense lamellar structures. These are significant biomedical materials that have been used for drug delivery applications in tissue engineering [8,11]. In bone tissue engineering, coral was found to have a close resemblance to trabecular bone owing to its porous structure and well-presented pore interconnectivity. However, there are two major demerits that limit its wide application in bone augmentation, including poor biomechanical properties and deprived osteoinductive behaviour. These poor biomechanical properties were addressed by altering coral skeletons into calcium phosphate-based coralline through hydrothermal process, which maintains the coral structure. This step can be achieved to improve its biomechanical properties using sol-gel coating. However, improving the osteoinductive properties of coral using current advancement remains at an early stage. In absence of this characteristic, the scaffold can only function as a carrier for the cells rather than supporting proliferation or inducing mesenchymal stromal cells differentiation into bone-like phenotype [12]. 

The incorporation of growth factors into the biomaterial scaffolds to support cell proliferation and osteogenic differentiation of mesenchymal stromal cells has been used as a strategy in bone tissue engineering. However, whether this strategy can be adopted in coral is yet to be investigated. In our preliminary study, it was proven that a specific growth factor, platelet-derived growth factor with BB variance (PDGF-BB), encapsulated in PLGA microsphere and incorporated in CORAGRAF, enhanced bone-marrow-derived mesenchymal stromal cell proliferation and osteogenic differentiation [13]. The PDGF-BB was the best selection due to its excellent capacity to induce mesenchymal stromal cell proliferation and differentiation compared to another variance of PDGF including CC, AA, DD and AB [14]. In the preclinical study, it was shown that PDGF-BB incorporated in different types of carriers played a significant role in bone formation. This was confirmed based on its role to induce osteoblastic differentiation, neo-collagen matrix formation, and bone mineral deposition [15]. On top of that, the appropriate carrier for a sustained release of growth factors to achieve minimum effective dose (MED) is important, otherwise the burst release may cause toxification to the cells [16]. Natural polymers, such as collagen, gelatine, and poly lactic-co-glycolic acid in combination with albumin and hyaluronic acid have been widely used in drug delivery systems due to their excellent biocompatibility when compared with synthetic carriers [17,18,19,20]. Collagen exhibits superior biocompatibility among natural polymers because of its extensive usage in clinical applications, favourable physical and degradation characteristics, and its excellent drug delivery system [21]. It was learned that poly (lactic-co-glycolic acid) (PLGA) was a Food and Drug Administration (FDA)-approved therapeutic device and an excellent option for the sustained release of drugs, owing to its biodegradability and biocompatibility properties. The sustained release of drugs to achieve MED during degradation of PLGA is the unique feature of this polymer, and thus it was highly adopted in surgical-associated procedures [22]. In the bone tissue engineering, the encapsulation of growth factors in PLGA microsphere to induce osteogenesis is a comment norm [23]. This was also evidenced in our previous study where the CORAGRAF (scaffold made from coral, which predominately contains calcium carbonate) loaded with PLGA encapsulated with PDGF-BB significantly induced osteogenic differentiation of seeded mesenchymal stromal cells (MSCs) in an in vitro condition [13]. The supply of MSCs together with this scaffold seems to be favourable for an optimal bone repair. The commitment of MSCs into an osteogenic phenotype can cause a paracrine effect that continuously induces native MSCs migration and osteogenic commitment at the defect region. This phenomenon can compensate for the limitation of growth factors including its short half-life. Therefore, the aim of the current study is to incorporate scaffolds with specific growth factors and MSCs to achieve excellent bone repair. Although our in vitro study has shown excellent outcomes, the mechanism of action of this 3D CORAGRAF + PLGA with PDGF-BB + MSCs in an in vivo condition mimicking clinical setting to repair critical bone defect is yet to be investigated elsewhere. 

In this study, the 3D CORAGRAF incorporated with PLGA microspheres loaded with PDGF-BB and seeded with MSCs was used to repair the calvaria critical-size bone defect created in a rat model. 

## 2. Results

### 2.1. Micro-CT Analysis

The repairing capacity of CORAGRAF was assessed based on continuous viewing of full-thickness defect region on weeks 4 and 8 using micro-CT imaging. The micro-CT images of week 4 and 8 confirmed that the defect size remained unchanged in terms of its size in the control group (Figure 1, Group I) without any treatment, and this attributed an excellent baseline control model for critical size defects. The group of rats that received at least CORAGRAF indicated a reduction in defect size as compared with the untreated control group (Figure 1, Groups II–IV). Most obviously, the group of rats that were treated with CORAGRAF incorporated with PLGA comprises PDGF-BB and seeded with MSCs indicated a larger radio-opaque mass, representing newly formed mineralized bony tissue from weeks 4 to 8 and almost a complete closure of defect was observed in week 8 (Figure 1, Group V). For a semi-quantitative analysis, the images of micro-CT were further used to calculate area of defect representing the level of regeneration occurring at the defect site (Figure 2). It was observed that there was a reduction of 4.5, 2 and 5.5-fold in week 4 and 1, 3.5 and 3.5-fold in week 8 of the defect area treated either with CORAGRAF, CORAGRAF + blank PLGA or CORAGRAF + PLGA with PDGF-BB, respectively. However, a significant reduction in the area of defect up to 8-fold (*p* = 0.008) in week 4 and 6-fold (*p* = 0.013) in week 8 was only observed when the defect was treated with CORAGRAF + PLGA with PDGF-BB + MSCs.

### 2.2. Histological Analysis of Bone Regeneration

The histological analysis was conducted to further support the micro-CT findings. A neo tissue formation with clear margin, as indicated in yellow dotted line, was observed in all the groups, except in rats from group I (Figure 3). In addition, a progression in tissue regeneration was evidenced when week 4 histological images were compared with images from week 8. Interestingly, the rats treated with CORAGRAF + PLGA with PDGF-BB + MSCs showed a significant tissue regeneration (complete osseous closure) at the defect region as per the histological images indicated (Figure 3, Group V). Within this regenerated tissue in group V, a blood vessel surrounded with new bone formation was observed. However, no inflammatory cells, multinucleated foreign body giant cells or tumours were observed in 8 weeks of observation in all groups treated with constructs. 

## 3. Discussion

In the current study, we investigated the reparative potential of CORAGRAF incorporated with PLGA comprises PDGF-BB and seeded with MSCs for critical bone defects in rat model. Five different groups were prepared, i.e., Group I: Untreated defect, Group II: CORAGRAF, Group III: CORAGRAF + blank PLGA, Group IV: CORAGRAF + PLGA with PDGF-BB and Group V: CORAGRAF + PLGA with PDGF-BB + MSCs for a parallel comparison. The untreated defect was used as a baseline control. 

A promising cell-based bone tissue engineering yields good outcomes when it involves a combination of cells, biomaterial scaffolds, and signalling molecules to restore the damaged bone region. The use of porous biomaterial can facilitate cell attachment, proliferation, and, at the later stage, formation of new tissue such as connective and vascular tissues [24]. Although some evolutions have been noticed in this field, several demerits are yet to be addressed. This includes a sustained release of signalling molecules to achieve minimum effective concentration (MEC) at the microenvironment where the scaffold is implanted for optimal cell attachment, proliferation, and osteogenic differentiation of mesenchymal stromal cells (MSCs) [25]. A study was therefore developed to investigate the CORAGRAF incorporated with PLGA comprises PDGF-BB seeded with MSCs. The reason behind the selection of PDGF is because of its excellent stimulation potential of canonical pathway in MSCs for its osteogenic differentiation commitment [26,27]. To our knowledge, this is a novel study investigating the efficacy of PDGF in combination with CORAGRAF and MSCs for bone tissue engineering. 

A sustained release of incorporated drugs, growth factors, cytokines, and chemokines requires an encapsulation using appropriate materials [28]. For this reason, PLGA was employed, owing to its excellent biodegradability and biocompatibility and approved therapeutic devices by the FDA [29]. PLGA, which is recognised as smart polymer or delivery system, allows penetration of water from the surface to the centre of the release device, which in turn activates the process of hydrolytic degradation and diffusion of the products encapsulated within the polymer matrix [30]. This will avoid the burst release of compound encapsulated and toxic induction at the implanted region. The sustained release of PDGF-BB incorporated in PLGA was demonstrated in our previous study [30].

An appropriate selection of materials is crucial as delivery system for the sufficient concentration of PDGF-BB in the implant site. Calcium carbonate-based scaffolds, which serve as regulating factors, were found to be one of the appropriate scaffolds for repairing critical bone defects in animal model [31]. This preference is due to its excellent features that mimic the characteristics of bones such as the mineral contents, topography, pores and porosity, and most importantly its biomechanical properties [32]. In the current study, coral graft (CORACRAF) was employed, owing to its features that are expected for bone substitutes. Interestingly, it was found that the diameter of the pores in CORAGRAF was approximately 400–500 µm, which is close to the pores found in cancellous bone. This feature is important to accelerate vascularization for neo bony ingrowth [33]. The vascularisation can support a rapid exchange of various factors, i.e., nutrients, metabolic wastes, gaseous and growth factors between local tissue and systemic circulation [34]. Furthermore, this will assist the migration of circulating bone precursors for effective bone repair [35]. This phenomenon was observed in the current findings where formation of neo-bony tissue was observed in all groups treated with CORAGRAF. Furthermore, the presence of CORAGRAF could have provided the base for migration and attachment of bone precursors and mature osteoblast from inherent bone marrow derived from adjacent cavities. In fact, the sustained released of PDGF incorporated in PLGA microspheres in CORAGRAF could have further enhanced this migration process either instantly from adjacent bone marrow, or from circulating bone precursors via neovascularisation at a later stage of the bone repair mechanism at the defect region [36]. However, the osteoconduction process seems to be insufficient with just CORAGRAF or in combination with PDGF as evidenced in our micro-CT and histology images. This could be because the PDGF that might have initiated the rapid migration of bone precursors at the early stage which is scarcely enough to develop neo-bone, particularly at the critical defect site. In addition, the exhausting of PDGF, as well as its short half-life, may only provide a provisional bone regeneration. As a surplus to CORAFGRAF and PDGF, the osteoblast precursor, mesenchymal stromal cells (MSCs), were incorporated to produce a viable construct and sustained release of various osteoinduction factors by MSCs in long run. This combination accomplishes the tissue engineering triad comprises cells, scaffold, and growth factors to boost neo-bone regeneration and speed up the healing process [37]. 

Furthermore, in pathophysiological conditions such as bony defects either in human or murine, the initial repair phase will be started with severe inflammation within one week where infiltration of neutrophil will be observed [38]. This process is followed by a moderate inflammation within three weeks where the presence of lymphocytes can be observed. Finally, mild to absent inflammation can be evidenced within nine weeks, with a gradual surge in collagenisation and new blood vessel formation. This phenomenon was clearly observed in the treatment group with CORAGRAF + PLGA with PDGF-BB + MSCs (Group v) where in 8 weeks, new bony ingrowth with vascularisation was initiated. This phenomenon was similar to in a previous article where the human bone cell seeded with biphasic ceramic scaffold promoted excellent bony ingrowth in rats with critical cranial defect [39]. The critical defect was almost completely closed by the osseous tissue as shown in micro-CT images of week 8 in this group. This approach was corroborated with another study where an excellent bony ingrowth was observed when the critical cranial defect in a rat model was treated with 3D spheroids, encapsulated in constructs of 3D-printed poly-L-lactide-co-trimethylene carbonate scaffolds and modified human platelet lysate hydrogels (PLATMC-HPLG) [40]. However, the advantage of using CORAGRAF can surpass any other studies in a similar approach solely due to its excellent biocompatibility and, most importantly, its provisional biomechanical support when the constructs are being implanted at the weight-bearing region. 

With regards to PDGF-BB, it was reported that the bone-marrow derived MSCs were unsuccessful to enter osteogenic lineage when treated with PDGF-BB at the concentration of 20 µg/mL [41]. However, this scenario was contradictory to our findings where the sustained release of PDGF-BB loaded at the concentration of 25 µg/mL in PLGA induced a successful osteogenic differentiation of MSCs in our in vitro culture conditions, as well as excellent bony ingrowth in an in vivo calvaria critical-size defect model in rats. Our findings were also corroborated with another study where significant angiogenesis and osteogenesis were observed in a calvarial critical-size defect model in rats [42]. This study adopted scaffold (porous calcium phosphate cement)-based cell delivery as similar to our study design in an in vivo model. This scaffold-based approach together with a sustained release of PDGF-BB could be the reason for the osteoinduction in MSCs. However, the mechanism of action of MSCs commitments to osteogenic lineage with the presence of PDGF-BB and scaffolds is worth to explore further. 

However, several issues, such as limited cell growth capacity in vitro, graft fixation techniques, long-term results in in vivo conditions, evaluation, and compliance with good manufacturing standards, must be assessed before considering this construct in clinical use [43].

## 4. Materials and Methods

### 4.1. Isolation and Culture of Rat Bone Marrow-Derived Mesenchymal Stromal Cells

Ethics approval was granted for this study from the Institutional Animal Care and Use Committee (IACUC), University of Malaya, Kuala Lumpur, Malaysia (Ethics number: 2017-180309/ORTHO/R/TKZ). The animal study was performed entirely at the Association for Assessment and Accreditation of Laboratory Animal Care International (AAALAC International, Frederick, MD, USA) accredited Animal Experimental Unit (AEU), Faculty of Medicine, University of Malaya, Kuala Lumpur, Malaysia. FIVE (5) rats were used for bone-marrow isolation. The rats were euthanized using CO_2_, and their lateral left and right hind limbs were shaved and draped with 10% povidone-iodine. A lateral longitudinal incision was made using a sharp blade to expose the left and right tibia and femur, and the tibias and femurs were separated from the surrounding muscles and tendons. The tibias and femurs were placed in a 50 mL polypropylene tube containing PBS (1X) (Gibco, Invitrogen, Waltham, MA, USA) and transferred to the tissue culture laboratory. In sterile conditions, the muscles and other tissues were scraped off using a sharp blade. Both of the ends of the bone were cut using a sharp bone cutter and the bone marrow was flushed out using a needle (26 gauge) with PBS (1X) in a 20 mL syringe. The purification of MSCs was performed via a standard Ficoll-Paque density gradient centrifugation (density 1.084 g/mL) as per the manufacturer’s instructions (GE Healthcare Bio-Sciences, St. Louis, MO, USA). The density gradient centrifugation was performed for 25 min at 2200 rpm. The interface containing mononuclear cells (MNCs) was isolated and washed thrice with PBS (1X). The cell culture was carried out in low glucose Dulbecco modified Eagles’ Medium (DMEM, Invitrogen, Waltham, MA, USA), supplemented with 10% fetal bovine serum (FBS, Invitrogen), 100 U/mL penicillin (Sigma-Aldrich, St. Louis, MO, USA) and 100 mg/mL streptomycin (Sigma-Aldrich, St. Louis, MO, USA). The number of cells and their viability was enumerated using Trypan blue exclusion method. Almost 1 × 10^6^ cells were seeded onto the T-75 culture and then incubated at 37 °C in 5% CO_2_ with 95% humidity. For the purpose of subsequent passaging, the cells in passage-0 (P0), having reached 80% confluency, were then washed using PBS (1X) and later incubated in trypsin (TrypLE, Gibco, Waltham, MA, USA) for 3 min in a CO_2_ incubator at 37 °C for complete cell detachment. The harvested P0 cells were sub-cultured again in passage-1 (P1) and the culture medium was changed every 72 h.

### 4.2. Scaffold Fabrication, Cell Seeding and Culture

The CORAGRAF (diameter: 5 mm × height: 5 mm, cylindrical scaffold), incorporated either with blank PLGA microspheres or PLGA microspheres contain PDGF-BB (25 μg/mL), was fabricated using our established methods [44]. At a density of 3.0 × 10^4^ cells/cm^2^, the MSCs were seeded onto the CORAGRAF + PLGA with PDGF-BB. Subsequently, the MSCs-seeded scaffolds (Group V) were used for transplantation. 

### 4.3. Creation of Calvaria Critical-Size Defect

A total of forty (*N* = 40) rats were used to create calvaria critical-size defect. Prior to surgery, the rats were anesthetized using 50 mg/kg ketamine and 5 mg/kg xylazine hydrochloride. Following anaesthesia, the surgical areas were shaved and disinfected with povidone-iodine. The surgery was performed by trained orthopaedic surgeons with the help of veterinarians. The rats were covered with a sterile surgical towel while exposing only the calvaria proximal–medial spot for incision. A vertical incision was made in the dimensions of 1.0–1.5 cm, and the soft tissue and the periosteum were elevated to expose the calvaria. A 5 mm in diameter calvaria critical-size defect (unilateral) was created using a micro drill at low rotation speed with constant irrigation. Repositioning and suturing of the periosteum were carried out with 5–0 PDS suture, while the skin was sutured using 4–0 silk suture. For the post-surgery management (pain management and prevention of infection), the rats were given an intraperitoneal injection of 15 mg/kg Kombitrim and 1mg/kg Meloxicam for three consecutive days. The rats were given food and water ad libitum during post-surgery recovery. The rats were monitored for 4 weeks after the post-surgery prior to use them for treatment. 

### 4.4. Transplantation of Scaffold

The rats were divided into five (5) groups (8 rats/group): Group I: Untreated defect, Group II: CORAGRAF, Group III: CORAGRAF + blank PLGA, Group IV: CORAGRAF + PLGA with PDGF-BB, and Group V: CORAGRAF + PLGA with PDGF-BB + MSCs. Prior to surgery, the rats were anaesthetized using 50 mg/kg ketamine and 5 mg/kg xylazine hydrochloride. Following anaesthesia, the surgical areas were shaved and disinfected with povidone-iodine. The rats were covered with sterile surgical towel while exposing only the calvaria proximal–medial spot of the previous scar. The defect site was opened through incision over the outline of the scar. The constructs were transplanted based on the composition of groups assigned, except group I without a scaffold (Figure 4A,B). For the post-surgery management, the rats were given an intraperitoneal injection of 15 mg/kg Kombitrim and 1 mg/kg Meloxicam for three consecutive days. The rats were given food and water ad libitum during post-surgery recovery. About four (*n* = 4) rats/time-point/group were sacrificed using overdose CO_2_ euthanasia method on weeks 4 and 8 prior to use for microcomputed tomography (micro-CT) imaging of the defect region and histological evaluations of the bone tissues harvested from defect margin of all groups. 

### 4.5. Microcomputed Tomography (Micro-CT)

The rats sacrificed at weeks 4 and 8 were subjected to micro-CT imaging. Calvaria specimens were evaluated using micro-CT scanner (Gamma Medical-Ideas, Northridge, CA, USA). The images had a slice thickness of 15 µm and a resolution of 2048 × 2048 pixels. The cylindrical region of interest (ROI) that corresponded to the calvaria defect was chosen using MicroView software. The images were captured and further processed to estimate bone regeneration in all groups based on size of defect (area) using ImageJ software (IJ 151j/Java 1.8.2-64-bit, NIH, Bethesda, MD, USA). 

### 4.6. Histological Assessment of Bone Regeneration Using Haematoxylin and Eosin

After the micro-CT images were taken, the skin and soft tissues on the skull were removed, and the harvested tissues from the calvaria proximal–medial spot were fixed in 10% neutral-buffered formalin overnight prior to being used for decalcification with 10% EDTA for 72 h. The decalcified tissues were embedded in paraffin and the tissue blocks were sectioned up to 5 µm thickness. The tissue sections were then rehydrated through a serial xylene/ethanol/water rehydration procedure before being stained with H&E using a standard procedure. The H&E-stained tissue sections were used to capture images with a bright field microscope (Nikon, Minato-ku, Tokyo).

### 4.7. Statistical Analysis

Statistical differences between groups were determined performing Mann–Whitney U and Kruskal–Wallis tests using SPSS version 25 (SPSS Inc., Chicago, IL, USA), or GraphPad prism version 8 statistical tools. The differences were considered statistically significant if the value of *p* was less than 0.05.

## 5. Conclusions

Encapsulation of MSCs in CORAGRAF together with PLGA comprises PDGF supported an excellent bony ingrowth in calvaria critical-size bone defect as compared with untreated rats or rats treated either only with CORAGRAF or in combination with PDGF. The micro-CT images indicated an enhanced bone mineralisation/ossification with almost complete closure of critical defects with neo-bone tissues within 8 weeks in rats treated with CORAGRAF + PLGA with PDGF-BB + MSCs. The histological findings confirmed the absence of immunogenicity, fibrosis, or tumour formation. Our previous findings from an in vitro characterisation, together with current in vivo outcomes, clearly indicated that CORAGRAF incorporated with PLGA comprises PDGF-BB for its sustained release to support MSCs ossification can be proposed for further assessment of its safety and efficacy in clinical trials. This will allow us to consider the use of this construct for widespread application in patients with critical bone defects.

## Figures and Tables

**Figure 1 marinedrugs-20-00561-f001:**
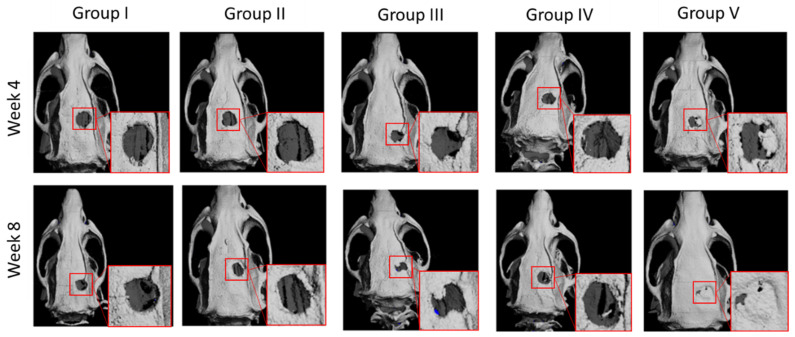
Reconstructed 3D micro-CT images of rat calvaria implanted with scaffolds after 4 and 8 weeks. Group I: Untreated defect, Group II: CORAGRAF, Group III: CORAGRAF + blank PLGA, Group IV: CORAGRAF + PLGA with PDGF-BB and Group V: CORAGRAF + PLGA with PDGF-BB + MSCs. The micro-CT images are best representative of at least three (3) biological replicates.

**Figure 2 marinedrugs-20-00561-f002:**
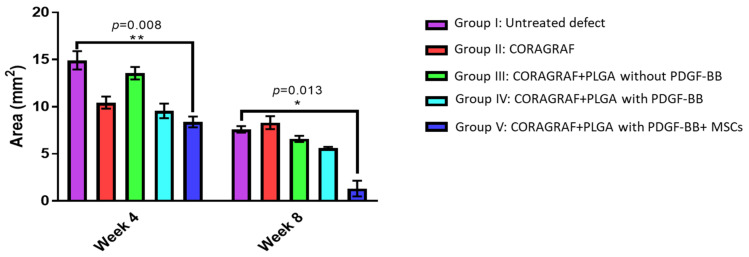
The area defect region on week 4 and 8 post-transplantation. The area of defect based on micro-CT micrographs was calculated using ImageJ software. The comparison of mean rank between groups and between week 4 and 8 of same group for all groups was performed using Kruskal–Wallis test and Mann–Whitney U test, respectively. Statistical significance * *p* < 0.05 and ** *p* < 0.01.

**Figure 3 marinedrugs-20-00561-f003:**
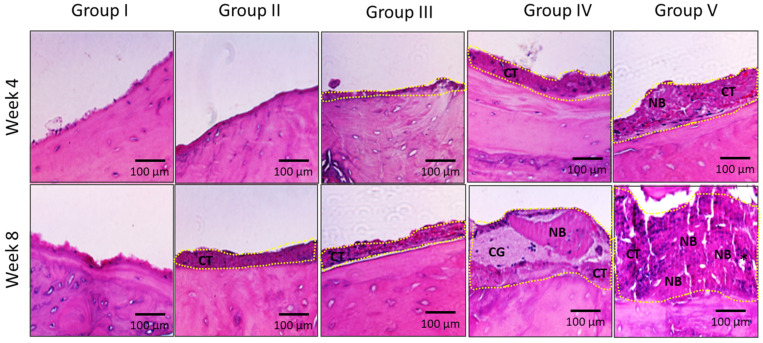
H&E analysis of calvaria proximal-medial spot treated with or without scaffold on week 4 and 8. Group I: Untreated defect, Group II: CORAGRAF, Group III: CORAGRAF + blank PLGA, Group IV: CORAGRAF + PLGA with PDGF-BB and Group V: CORAGRAF + PLGA with PDGF-BB + MSCs. The micro-CT images are best representative of at least three (3) biological replicates. The yellow dotted line indicates the margin of neo bony-like tissue formation. Pictures show the osteo-differentiation process, with new bone (NB), blood vessels (*asterisks*), connective tissue (CT) and CORAGRAF (CG).

**Figure 4 marinedrugs-20-00561-f004:**
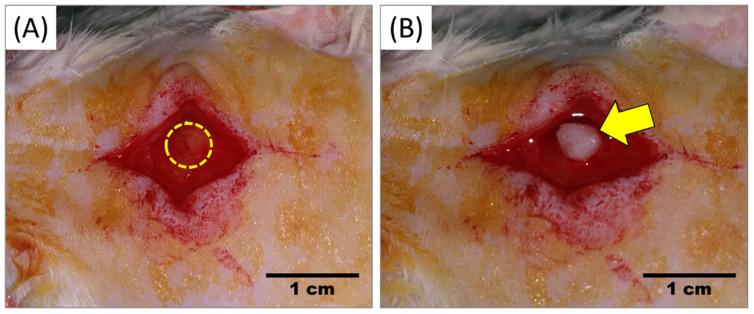
After fur was shaved and disinfected with povidone-iodine, skin incision was made over the outline of scar. (**A**) Untreated defect (control) and (**B**) Defect treated with construct (CORAGRAF + PLGA with PDGF-BB + MSCs). Skin was then closed with suture. Dotted outline: Defect outline and Arrow: defect treated with construct.

## Data Availability

The authors confirm that the data supporting the findings of this study are available within the article.

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
