# Peer review of "Potential Use of 3D CORAGRAF-Loaded PDGF-BB in PLGA Microsphere Seeded Mesenchymal Stromal Cells in Enhancing the Repair of Calvaria Critical-Size Bone Defect in Rat Model"

_marinedrugs, 2022, doi:10.3390/md20090561_

Round 1
Reviewer 1 Report
Lines 187-189: “Figure 1 (...) (B) either of CORAGRAF, CORAGRAF+blank PLGA, CORAGRAF+PLGA with PDGF-BB and CORAGRAF+ PLGA with PDGF-BB+MSCs.”
1 - It is not logical that figure “1B” expresses all groups in a single image. Is it possible for the authors to redo the figure in order to contain the macroscopic aspect of implantation of each of the 4 experimental groups, in addition to the empty defect (Fig.1A)? If this is not possible, identify at least which implanted group is really represented in Fig.1B.
Lines 214-224 and 244-258:
2 - Nothing was described in the microtomographic or histological analysis regarding the residual presence or absence of biomaterials from the experimental times of 4 or 8 weeks. Did the computed tomography consider this possible bias, to distinguish the pattern of the newly formed bone from the crystalline aspect of the Coragraft? Were no fragments compatible with Coragraft or PLGA detected in the histological analysis?
Line 244-258: The histological technique and analysis contain many weaknesses and need further revision.
3 - It is necessary for the authors to specify the location of the images represented, considering the edge-to-edge path of the critical bone defect. The images suggest that they correspond to different areas of the edge of the defect and the center of the defect, in addition to not following a homogeneous presentation pattern among themselves. The intention for choosing this random representation is not clear.
4 - The quality of the images is not good enough to warrant the authors' claim that there was a lot of neoformed bone tissue in addition to the native bone. In order for the reader to understand the results, a more in-depth description of the expected morphology of neoformed bone tissue in the groups is necessary, including: areas of greater cellularity (highlighting the Group V), process of morphological change of cells in osteodifferentiation (typical fibroblast-like appearance of mesenchymal cells or cuboidal similar to osteoblasts to the flattened and dendritic osteocyte-like pattern), the deposition between osteocytes of eosinophilic collagenous osteoid matrix and the presence of active osteoblasts in a monolayer at the edge of the tissue. Suggested reading that shows the morphology of osteodifferentiation of implanted ceramic groups (with and without mesenchymal cells) in rat cranial defects: Lomelino Rde O, Castro-Silva II, Linhares AB, Alves GG, Santos SR, Gameiro VS, Rossi AM, Granjeiro JM. The association of human primary bone cells with biphasic calcium phosphate (βTCP/HA 70:30) granules increases bone repair. J Mater Sci Mater Med. 2012 Mar;23(3):781-8. doi: 10.1007/s10856-011-4530-1. Epub 2011 Dec 27. PMID: 22201029.
5 - Within the biological responses in the host, did the authors identify, in the slides presented, a mixture of inflammatory cells (mono or polymorphonuclear) or a fibrous capsule with a foreign body reaction, along the path of the bone defect? If not, describe these parameters, in addition to the aforementioned “no tumor formation” (Line 350). It is important to describe both non-irritation or immunogenicity and non-fibrogenesis promoted by the groups, within tissue engineering proposals aimed at bone regeneration.
6 - For the authors to interpret that there was superiority between treatments or significant results, it is necessary to use some quantitative histomorphometric technique, more reliable than microtomography (which only evaluates mineralized bone and not the organic matrix or osteoid). Was there any histological measurement, by point or area, or semi-quantitative analysis by scores? Suggested reading that shows histomorphometry of polymeric biomaterials or natural composites implanted in rat cranial defects: Souza FFP, Pérez-Guerrero JA, Gomes MJP, Cavalcante FL, Souza Filho MSM, Castro-Silva II. Development and characterization of poultry collagen-based hybrid hydrogels for bone regeneration. Acta Cir Bras. 2022 May 13;37(3):e370302. doi: 10.1590/acb370302. PMID: 35584534; PMCID: PMC9109989.
Line 259-345:
7 - There was a discussion based on poor evidence, not considering the performance (or amount of generated bone) against the use of other natural biomaterials, associated or not with growth factors and cells. It is worth going through the suggested references, to make this section of the article more robust.
Author Response
|
Reviewer 1 Queries |
||
|
Queries |
Authors Response |
|
|
Lines 187-189: “Figure 1 (...) (B) either of CORAGRAF, CORAGRAF+blank PLGA, CORAGRAF+PLGA with PDGF-BB and CORAGRAF+ PLGA with PDGF-BB+MSCs.” 1 - It is not logical that figure “1B” expresses all groups in a single image. Is it possible for the authors to redo the figure in order to contain the macroscopic aspect of implantation of each of the 4 experimental groups, in addition to the empty defect (Fig.1A)? If this is not possible, identify at least which implanted group is really represented in Fig.1B. |
Thanks for the reviewer’s comments. The picture with construct for all group looks identical from eyeballing. For this reason, we used the CORAGRAF+ PLGA with PDGF-BB+MSCs which is our ultimate testing group for representative in Figure 1B. Otherwise it seems like repetition. We follow the review’s suggestion to state the representative group of picture 1B. Correction has been done (Line 187, Pg 7).
|
|
|
Lines 214-224 and 244-258: 2 - Nothing was described in the microtomographic or histological analysis regarding the residual presence or absence of biomaterials from the experimental times of 4 or 8 weeks. Did the computed tomography consider this possible bias, to distinguish the pattern of the newly formed bone from the crystalline aspect of the Coragraft? Were no fragments compatible with Coragraft or PLGA detected in the histological analysis? |
Thanks for the reviewer’s comment. I totally agree with reviewer that the presence of residual can be a potential bias without a clear regenerative potential of construct studied. This is the reason where a comparison was made between week 4 and week 8 side by side to rule out the residual bias. Although the µCT is incapable to distinguish between biomaterial residual and neo bony regeneration, the week 8 outcomes as compared with week 4 clearly indicate the neo bony regeneration in our study.
|
|
|
Line 244-258: The histological technique and analysis contain many weaknesses and need further revision. 3 - It is necessary for the authors to specify the location of the images represented, considering the edge-to-edge path of the critical bone defect. The images suggest that they correspond to different areas of the edge of the defect and the center of the defect, in addition to not following a homogeneous presentation pattern among themselves. The intention for choosing this random representation is not clear.
|
Thanks for reviewer’s excellent comment. This comment really helps us to ponder the way we should explain the location of the defect that represent the microscopic evidence of bone regeneration.
Based on our recall, the images were taken from 3-9 O‘clock range. This was our best location and space we used to display the bony regeneration using histology.
To be honest, one of the greater technical challenges that we faced was to get a good imaging from the samples that have been treated with decalcifying agent. However, we still managed to cover area between 3-9 O‘clock range for presentation. |
|
|
4 - The quality of the images is not good enough to warrant the authors' claim that there was a lot of neoformed bone tissue in addition to the native bone. In order for the reader to understand the results, a more in-depth description of the expected morphology of neoformed bone tissue in the groups is necessary, including: areas of greater cellularity (highlighting the Group V), process of morphological change of cells in osteodifferentiation (typical fibroblast-like appearance of mesenchymal cells or cuboidal similar to osteoblasts to the flattened and dendritic osteocyte-like pattern), the deposition between osteocytes of eosinophilic collagenous osteoid matrix and the presence of active osteoblasts in a monolayer at the edge of the tissue. Suggested reading that shows the morphology of osteodifferentiation of implanted ceramic groups (with and without mesenchymal cells) in rat cranial defects: Lomelino Rde O, Castro-Silva II, Linhares AB, Alves GG, Santos SR, Gameiro VS, Rossi AM, Granjeiro JM. The association of human primary bone cells with biphasic calcium phosphate (βTCP/HA 70:30) granules increases bone repair. J Mater Sci Mater Med. 2012 Mar;23(3):781-8. doi: 10.1007/s10856-011-4530-1. Epub 2011 Dec 27. PMID: 22201029. |
Thanks to the reviewer for the great suggestion about an in-depth analysis on the histological findings.
Based on the histological findings refereeing to the article suggested, the new bone and new blood vessel formation have been clearly indicated in the Figure 4 with the annotation in the figure legend (Line 243-250, pg.7).
The result (Line 257-260, pg. 7) and discussion (Line 327-332, pg. 8) have been updated with index citation of suggested article.
The only limitation is to observe the histology images with high mag. to distinguish between new bone cells from resident cells as image quality is compromised. With your permission, we will consider this suggestion during our study using large animal model to test this construct for FDA compliance. |
|
|
5 - Within the biological responses in the host, did the authors identify, in the slides presented, a mixture of inflammatory cells (mono or polymorphonuclear) or a fibrous capsule with a foreign body reaction, along the path of the bone defect? If not, describe these parameters, in addition to the aforementioned “no tumor formation” (Line 350). It is important to describe both non-irritation or immunogenicity and non-fibrogenesis promoted by the groups, within tissue engineering proposals aimed at bone regeneration. |
Thanks to the reviewer’s comment for highlighting this point. There were no inflammatory cells or multinucleated foreign body giant cells observed in the histological findings. This statement has been included in the results (Line 259-260, pg. 7) and conclusion statements (Line 254-255, pg. 9) |
|
|
6 - For the authors to interpret that there was superiority between treatments or significant results, it is necessary to use some quantitative histomorphometric technique, more reliable than microtomography (which only evaluates mineralized bone and not the organic matrix or osteoid). Was there any histological measurement, by point or area, or semi-quantitative analysis by scores? Suggested reading that shows histomorphometry of polymeric biomaterials or natural composites implanted in rat cranial defects: Souza FFP, Pérez-Guerrero JA, Gomes MJP, Cavalcante FL, Souza Filho MSM, Castro-Silva II. Development and characterization of poultry collagen-based hybrid hydrogels for bone regeneration. Acta Cir Bras. 2022 May 13;37(3):e370302. doi: 10.1590/acb370302. PMID: 35584534; PMCID: PMC9109989. |
Thanks for the reviewer’s suggestion. It would excellent if the data is presented with histomorphometric analysis. However due to a limitation in technical aspect such as the absence of digital scanner to produce images to produce a contiguous field covering the extension of the bone defect, the qualitative analysis was performed to display the outcome to our best. Our sincere apology for the limitation in this sense, however we will try to outsource the technique if available when we are planning to test the construct on large animal where the ethics application is underway.
|
|
|
Line 259-345: 7 - There was a discussion based on poor evidence, not considering the performance (or amount of generated bone) against the use of other natural biomaterials, associated or not with growth factors and cells. It is worth going through the suggested references, to make this section of the article more robust.
|
Thanks for the reviewer’s comment.
The discussion part was improved especially from Pg 321-333 to make it more supportive with citing references as stated above. |
|
Reviewer 2 Report
This manuscript describes the transplantation of mesenchymal stromal cells (MSCs) with scaffolds and PDGF-BB-loaded microspheres in bone defected rats. The authors showed that MSCs with both the scaffolds and PGFF-BB loaded microspheres successfully enhance the healing of bone defects, which were judged from the data such as the closure of bone defects and H&E staining. Although obtained results were persuasive, the manuscript needed some improvement. Therefore, the reviewer recommends that the authors should be accepted after the following change:
- The descriptions regarding the term “CORAGRAF” were not provided throughout the manuscript. The authors should provide the definition and explanation of “CORAGRAF”.
- The caption of Figure 3 showed that the comparison of means was carried out by using Mann-Whitney U and Kruskal-Wallis tests, which were for between two groups and more than three groups, respectively. However, the reviewer cannot recognize which tests were carried out for the comparison of means among five groups in Figure 3. The authors must perform appropriate statistic test, and provide which test was performed in Figure caption.
- Detailed information about transplanted scaffolds such as the dimension, structure, and size, and the formulations of PLGA microspheres should be provided in the manuscript.
Author Response
|
Reviewer 2 Queries |
||
|
8 |
The descriptions regarding the term “CORAGRAF” were not provided throughout the manuscript. The authors should provide the definition and explanation of “CORAGRAF”. |
Thanks to the reviewer’s suggestion. CORAGRAF is a scaffold derived from coral that predominantly contains calcium carbonate. This statement was include in the manuscript (Line 103, pg. 3) |
|
|
The caption of Figure 3 showed that the comparison of means was carried out by using Mann-Whitney U and Kruskal-Wallis tests, which were for between two groups and more than three groups, respectively. However, the reviewer cannot recognize which tests were carried out for the comparison of means among five groups in Figure 3. The authors must perform appropriate statistic test, and provide which test was performed in Figure caption. |
Thanks to the reviewer’s highlight. The figure 3 caption was improved with clearly mentioning of the groups and its respective statistical analysis (Line 222-224, pg. 6). |
|
|
Detailed information about transplanted scaffolds such as the dimension, structure, and size, and the formulations of PLGA microspheres should be provided in the manuscript. |
Thanks for the reviewer’s suggestion. Since this is a continuation of previous in vitro study, therefore the detailed information of construct preparation has been cited in the method to avoid infringement.
However, some basic information has been provided to facilitate the readers (Line 148-150, Pg 3).
We are also happy to provide the detailed information in the supplementary file if the reviewer prefers that way.
|